# Control Measures for SARS-CoV-2: A Review on Light-Based Inactivation of Single-Stranded RNA Viruses

**DOI:** 10.3390/pathogens9090737

**Published:** 2020-09-08

**Authors:** Joshua Hadi, Magdalena Dunowska, Shuyan Wu, Gale Brightwell

**Affiliations:** 1AgResearch Ltd., Hopkirk Research Institute, Cnr University Ave and Library Road, Massey University, Palmerston North 4442, New Zealand; Joshua.Hadi@agresearch.co.nz (J.H.); Shuyan.Wu@agresearch.co.nz (S.W.); 2School of Veterinary Science, Massey University Manawatu (Turitea) Tennent Drive, Palmerston North 4474, New Zealand; M.dunowska@massey.ac.nz; 3New Zealand Food Safety Science and Research Centre, Massey University Manawatu (Turitea) Tennent Drive, Palmerston North 4474, New Zealand

**Keywords:** SARS-CoV-2, single-stranded RNA viruses, viral inactivation, ultraviolet, blue light, red light

## Abstract

SARS-CoV-2 is a single-stranded RNA virus classified in the family *Coronaviridae*. In this review, we summarize the literature on light-based (UV, blue, and red lights) sanitization methods for the inactivation of ssRNA viruses in different matrixes (air, liquid, and solid). The rate of inactivation of ssRNA viruses in liquid was higher than in air, whereas inactivation on solid surfaces varied with the type of surface. The efficacy of light-based inactivation was reduced by the presence of absorptive materials. Several technologies can be used to deliver light, including mercury lamp (conventional UV), excimer lamp (UV), pulsed-light, and light-emitting diode (LED). Pulsed-light technologies could inactivate viruses more quickly than conventional UV-C lamps. Large-scale use of germicidal LED is dependent on future improvements in their energy efficiency. Blue light possesses virucidal potential in the presence of exogenous photosensitizers, although femtosecond laser (ultrashort pulses) can be used to circumvent the need for photosensitizers. Red light can be combined with methylene blue for application in medical settings, especially for sanitization of blood products. Future modelling studies are required to establish clearer parameters for assessing susceptibility of viruses to light-based inactivation. There is considerable scope for improvement in the current germicidal light-based technologies and practices.

## 1. Introduction

Three coronaviruses are known to cause severe pneumonia in humans, namely severe acute respiratory syndrome coronavirus (SARS-CoV) [1,2], middle east respiratory syndrome coronavirus (MERS-CoV) [3], and SARS-CoV-2 [4,5]. Four other coronaviruses are associated with common cold symptoms: human coronavirus (HCoV) 229E, HCoV NL63, HCoV OC43, and HCoV HKU1, which complete the group of seven coronaviruses known to infect humans [6]. Infections with SARS-CoV and MERS-CoV have resulted in approximately 10,000 cumulative cases of pneumonia, with death rates of 9.6% and 34.4%, respectively [7,8]. On 7 January 2020, the World Health Organization (WHO) identified a novel coronavirus SARS-CoV-2 in the Wuhan province of China [9], which has since caused a worldwide pandemic, with more than 21.2 million confirmed cases and over 760,200 confirmed deaths as of 14 August 2020 (https://www.worldometers.info/coronavirus/).

The precise infection routes of SARS-CoV-2 remain undefined, although respiratory droplets are considered as the main mode of transmission and thus, it has been recommended that maintaining a safe distance of at least 1 m from people with an unknown infection status will minimize the risk of SARS-CoV-2 infection [10]. However, accumulating evidence suggests that respiratory droplets can be airborne and travel for up to 8 m under some circumstances [11,12,13,14,15,16,17,18,19,20]. These findings are consistent with a study that reported on the detection of SARS-CoV-2 (viral RNA) in the air approximately 4 m from infected patients in a hospital ward [21]. Two in vitro studies also found that SARS-CoV-2 retained its infectivity in aerosols for 3 h [22] and 16 h [23].

In several studies, SARS-CoV-2 was detected on solid surfaces and in liquid media. Firstly, the virus has been identified in stool samples from asymptomatic and symptomatic patients by real-time reverse-transcriptase polymerase chain reaction (RT-PCR) [24,25,26], including live SARS-CoV-2 (four SARS-CoV-2 positive samples were cultured and live virus was detected by electron microscopy) [25]. However, there is no conclusive evidence on whether SARS-CoV-2 retains its infectivity after passing through the digestive system. Further, SARS-CoV-2 (viral RNA) has also been detected in untreated wastewater [27,28]. Under experimental conditions, SARS-CoV-2 was found to remain infectious after being deposited on surfaces, such as steel, plastic, and copper, for hours to days [22].

These findings highlight the potential importance of environmental contaminations (air, liquid, and solid surfaces) for the dissemination of SARS-CoV-2 in the population, especially given that undetected viral shedding by asymptomatic, pre-symptomatic (viral shedding during the incubation period), or mildly symptomatic individuals is relatively prevalent [29,30,31,32,33]. This warrants the development and implementation of control measures to reduce the viability of SARS-CoV-2 in the environment, especially interventions that could be applicable for routine use in public places, such as airports, shopping centers, office spaces, and others. A review by Aboubakr, Sharafeldin, and Goyal (2020) summarized the current knowledge of the persistence of coronaviruses, including SARS-CoV-2, in the environment (air, liquid, and solid) at different temperatures and under varying levels of relative humidity [34].

Light-based technologies have been extensively used for their antimicrobial properties [35,36,37,38]. However, relatively little data are available on the virucidal effects of light treatment. Thus, the purpose of this review was to assess the effects of light treatment on the viability of coronaviruses, especially SARS-CoV-2. Due to the limited data available on the photo-inactivation of coronaviruses, we also included data on the effects of light-based treatments on other single-stranded RNA (ssRNA) viruses. Finally, we have also discussed the potential mechanisms of light-based inactivation of SARS-CoV-2 and other ssRNA viruses (As shown in Figure 1).

## 2. Virion Structure of SARS-CoV-2

Severe acute respiratory syndrome coronavirus 2 (SARS-CoV-2) is an enveloped, single-stranded, and positive-sense RNA virus classified in the family *Coronaviridae* [39]. The genome of SARS-CoV-2 is between 29.8 to 29.9 kb in size [40,41,42] and has a total content of guanine and cytosine (G + C) of 38.02% [41], which is typical of coronaviruses [43], but places it at the lower end among RNA viruses (30–60%) [44].

The viral genome of coronaviruses contains at least six open reading frames (ORFs) [45] that encode replicase polyproteins, lipid-associated glycoproteins (spike (S), envelope (E), and membrane (M) proteins), nucleocapsid proteins (N), and other small proteins with unknown functions [46]. Four of these proteins (S, E, M, and N) are important to obtain a structurally-complete viral particle (Figure 2) [47,48].

As with other coronaviruses, the S-protein of SARS-CoV-2 consists of two functional subunits that are responsible for binding to the host receptor (S1) and facilitating fusion between the viral particle and the host cell membrane (S2) [49,50,51,52]. Six amino acid residues in the receptor-binding domain (RBD) are critical for the binding of S1 subunit to angiotengsin-converting enzyme 2 (ACE2) [53], although the complete binding mechanism has not been fully elucidated [54]. The compatibility of RBD residues with human ACE2 is a determinant factor for human-to-human transmission [55]. A study showed that a human/civet SARS-CoV isolate (isolated in 2003; GenBank accession number AY525636; strain GD03) exhibited no transmission between humans due to its lower affinity to human ACE2, as compared with two other SARS-related coronavirus isolates, namely SARS-CoV-2 (isolated in 2019; GenBank accession number MN908947.1) and human SARS-CoV (isolated in 2002; GenBank accession number NC_004718.3; strain Tor2), that were capable of inter-human transmission [55].

During an infection, the E-protein is mainly distributed in the endoplasmic reticulum and golgi apparatus of the host cell and participates in virus assembly, budding, morphogenesis, and trafficking [56,57,58,59]. The carboxyl-terminal of SARS-CoV E-protein interacts with proteins associated with Lin Seven 1 (PALS1), which is a member of the tight junction complex in the epithelial cells, and induces the redistribution of PALS1 [60,61]. This disrupts the formation of the tight junctions and thus contributes to the damage of the lung epithelium observed in SARS-infected patients [60,61]. Recombinant coronaviruses that lacked E-protein had significantly lower virus titres, impaired viral maturation, and lost their propagation ability in vitro [62,63,64,65].

The M-protein is the most abundant glycoprotein in the coronavirus virion and functions as a central organizer of the virus assembly [47,66]. The dimer conformation of the M-protein has two main roles: (1) promotes membrane curvature, which facilitates the budding and release of the virus particle, in association with the E-protein; (2) stabilizes both the internal core of the virion and the nucleocapsid [47,67,68,69]. Interactions between M- and S-proteins ensure that the S-protein is retained in the golgi or the endoplasmic reticulum/golgi intermediate compartment (ERGIC), where the virus assembles [70].

Coronavirus N-protein is a multifunctional protein that binds to the viral RNA to form ribonucleocapsid and provides protection to the viral genome [71]. The N-protein has been shown to assist with the genome packaging process, in association with the M-protein [72,73]. Moreover, the N-protein of SARS-CoV may facilitate immune evasion by suppressing RNA interference (RNAi) [74] and antagonizing the induction of type I interferon (IFN) in the host cells [75]. Other functions of N-protein have also been proposed, which include potential roles in virus replication, transcription, and translation [71].

The S-, E-, and M-proteins are associated with the viral lipid envelope [71] and thus, a disruption of the envelope may restrict the functions of these proteins. Similarly, the change in the viral capsid may render the viral genome susceptible to degradation, especially in the presence of ribonuclease enzymes. Damage to the viral genome may alter the expression of the viral proteins and thus, potentially inhibits the replication of the virus. In subsequent sections, we present discussions on several light-based sanitization techniques that may target these viral structures, which include analyses of several theories on the correlation between genomic properties and susceptibility to inactivation by light.

## 3. Viral Properties of Ultraviolet Light

### 3.1. Mechanism

Generally, the ultraviolet spectrum ranges from 10 to 380 nm and is divided into three regions: ultraviolet (UV; 300–400 nm), deep-ultraviolet (DUV; 200–300 nm), and far-ultraviolet (FUV; ≤200 nm) [76,77]. However, biologists have defined three types of UV radiation, namely UV-A (320–400 nm), UV-B (290–320 nm), and UV-C (200–290 nm), based upon its effects on living creatures, which excludes the FUV region of the UV spectrum [76]. Nevertheless, FUV, which is also known as vacuum ultraviolet (VUV), possesses a higher photon energy (6.70 eV) than other UV lights at longer wavelengths (4.88 eV at 254 nm and 3.40 eV at 365 nm) [78]. Thus, FUV readily breaks down chemical bonds to generate reactive oxygen species (ROS) and hydroxyl radicals. In this process, ozone is also generated, which limits the use of FUV due to its potential health risks [78].

Among the three types of UV radiation, antimicrobial activities have been mostly observed in the UV-C range at 253.7 nm [79,80] (maximum absorption of nucleic acids at 265 nm), which is commonly delivered via germicidal lamps. However, exposure to these lamps has been associated with health risks, which mainly involve damage to the eyes and skin [81,82]. Nevertheless, recent evidences suggested that UV-C at 222 nm exhibited germicidal activity [83,84,85], but inflicted no damage on the eyes and skin of mice [86,87,88]. These data are still preliminary and further research is needed to ascertain the safety of UV light at 222 nm, especially its potential long-term effects on human health.

UV-C and UV-B can be absorbed by DNA or RNA molecules and cause damage by inducing photo-chemical fusion of two adjacent pyrimidines into covalent-linked dimers [37,89]—racil/cytosine dimers in RNA and thymine/cytosine dimers in DNA [80]. There are six possible dimers that could be formed by UV-light treatments: (1) thymine/thymine; (2) cytosine/cytosine; (3) cytosine/thymine; (4) uracil/uracil; (5) thymine/cytosine; (6) uracil/cytosine [90]. Other mechanisms of UV-light inactivation include RNA-protein cross-linking [91] and site-specific damage to RNA or proteins through energy transfer between the two molecules [92].

Although the absorbance of UV-A by DNA is low, it can still induce antimicrobial inactivation via oxidative damage, such as nucleotide base damage or direct DNA strand breaks, in the presence of photosensitizers [89]. The endogenous photosensitizing pigments may be excited by the UV light, which results in the production of ROS, such as hydrogen peroxide. Further, UV light could also induce membrane damage via the oxidation of unsaturated fatty acids to hydrogen peroxides [37].

Inactivation mechanisms may vary across different viruses. For instance, genomic damage is the primary UV-light inactivation mechanism in MS2 bacteriophage (ssRNA virus) across the germicidal UV spectrum [93]. In adenoviruses (dsDNA virus), DNA damage is only responsible for the loss of infectivity at UV wavelengths above 240 nm, whereas damage to other viral components contributes to the loss of infectivity at UV wavelengths below 240 nm [94]. Similarly, the inactivation rate of Tulane virus (calicivirus with ssRNA) was found to be the same at 220 nm and 254 nm, whereas porcine rotavirus (OSU strain; dsRNA) was more sensitive to UV light at 220 nm than at 254 nm [95]. Further, UV-light treatment on Tulane virus resulted in damage to both its capsid binding protein and genome, whereas the porcine rotavirus was only inactivated via mutagenesis of its genome [95]. These findings should prompt researchers to be prudent in selecting viral surrogates—MS2 bacteriophage is a common surrogate for enteric viruses in the validation studies of viral clearance in samples treated by polychromatic UV reactor, despite the fact that enteric viruses may have different UV spectral sensitivities [93,96].

Theoretically, properties of the viral genome may control their susceptibility to light inactivation: (1) viruses with larger genomes would be more susceptible to light inactivation due to their larger target size; (2) viruses with double-stranded (dsRNA or dsDNA) genome may be more resistant due to the possibility of repair by the undamaged strand [97]. However, a modelling study found that there was no clear correlation between genome size (including single-stranded vs. double-stranded) or physical size, and susceptibility to UV-light inactivation [44].

Matallana-Surget et al. (2008) suggested that there was a strong correlation between the formation of cytosine-containing photoproducts (induced by UV-B) and increasing G + C contents (28–75%) in bacterial DNA [98]. However, Kowalski et al. (2009) found a moderate negative correlation (R^2^ = 0.46) between UV-light inactivation rate constant and G + C content across 28 ssRNA viruses [44]. Nevertheless, G + C content values alone could not be used as a sole predictor of UV susceptibility [44].

### 3.2. Inactivation of Viruses in Air

Viral RNA was reported in air samples from different areas of hospitals that treated SARS-CoV-2 patients [21,99,100], which implies that nosocomial transmission over distances beyond 1 to 2 m is possible, especially in settings in which aerosol-generating medical procedures are performed [10]. However, the presence of infectious SARS-CoV-2 in the air has only been confirmed in vitro [22,23] and thus, the global epidemiology of SARS-CoV-2 remains undefined.

There are limited data on the inactivation of airborne viruses, although several studies have assessed the inactivation of other airborne nosocomial pathogens, such as *Mycobacterium tuberculosis*, by ultraviolet germidical irradiation (UVGI). Two studies found that the presence of upper-room UV at 1.8 m or 2.1 m in hospital wards reduced airborne tuberculosis (TB) transmission, as observed from the significantly (*p* < 0.05) lower tuberculin skin test conversion or culturable *M. tuberculosis* in organs of guinea pigs that were exposed to UV-treated exhaust air from wards containing tuberculous patients than those in the control group (exposed to untreated air) [101,102]. The UV treatment at 2.3 m also lowered the concentration of culturable airborne bacteria, such as *Bacillus subtilis* spores (46–80% reduction), *Mycobacterium parafortuitum* (83–98%), and *Mycobacterium bovis* BCG (96–97%), in the breathable zone (1.5 m) [103]. However, despite its potential benefits, the use of UVGI has also been associated with certain health risks, such as photo-keratoconjunctivitis or photo-dermatitis [104], and thus must be safely installed within mechanical ventilation systems or in upper-room zones, possibly in combination with the use of louvers, to avoid direct exposure of unprotected eyes and skin to the UV light [82,104,105].

Several authors investigated the use of UV-based treatment to inactivate aerosolized ssRNA viruses (Table 1). From five studies, we calculated the average dosage of UV light required for achieving 90% inactivation of aerosolized ssRNA viruses (D_90_) of 0.71 mJ/cm^2^ (Table 4), although the varying experimental conditions have rendered a direct comparison between these studies to be inappropriate. There is no clear pattern that indicates different sensitivities to UV light across different types of ssRNA viruses, although a study found that single-stranded viruses (ssRNA and ssDNA) were more susceptible than double-stranded viruses (dsRNA and dsDNA) [97]. However, further research is required to ascertain the effects of UV light on viruses with different capsid types, genome size, and outer structures (enveloped and non-enveloped).

Higher relative humidity (RH) decreased the effectiveness of UV-based inactivation of airborne viruses [97,106,107], possibly due to the sorption of water onto the virus surface, which provided protection against DNA or RNA breakages [97]. A study also suggested that UV-light inactivation of airborne porcine reproductive and respiratory syndrome virus (PRRSV) was significantly (*p* = 0.0167) higher at lower temperatures (≤15 °C) than at higher temperatures (≥30 °C), although RH was still a more prominent factor—the virus was most susceptible, moderately susceptible, and least susceptible to UV-light inactivation at RH levels of 25–79%, ≤24%, and ≥80%, respectively [106]. However, the relationship between RH and temperatures, in regards to UV-light inactivation of airborne viruses, is not fully elucidated [106]. Several studies also found that human airway mucus supported prolonged survival of respiratory viruses, such as influenza A and influenza B viruses, in aerosols or droplets exposed to a range of RH [108,109,110], albeit no data are available on the effects of airway mucosal components on UV-light inactivation of these viruses. Thus, future studies are needed to investigate the efficacy of UV-light sanitization in matrixes that resemble the human airway surface liquid produced during a viral infection (aerosolized or stationary).

Most studies used the conventional low-pressure, mercury-vapour lamp (254 nm) due to their electrical efficiency (in terms of conversion of electrical power input to UV germicidal output) and low cost. However, concerns have been raised about the potential environmental and health effects of mercury, especially in the event of leakage [111]. There are alternative technologies, such as excimer lamps and light-emitting diode (LED), which also provide a wider selection of wavelengths. 

Excimer lamps emit UV light at different wavelengths based on the combination of gases within their dielectric materials [111]. For instance, xenon-bromine (285 nm) and krypton-bromine (207 nm) excimer lamps exhibited bactericidal activities against *Escherichia coli* [112] and methicillin-resistant *Staphylococcus aureus* [113], respectively. In studies on ssRNA viruses, krypton-chlorine (KrCl) excimer lamps (222 nm) were shown to inactivate aerosolized influenza A virus [84] and human coronaviruses [85].

For influenza A (H1N1), Welch et al. (2018) [84] observed that inactivation rates of the virus by KrCl excimer lamps (D_90_ = 1.28 mJ/cm^2^; D_95_ = 1.6 mJ/cm^2^) were comparable to the values (D_90_ = 1.04 mJ/cm^2^; D_95_ = 1.1 mJ/cm^2^) reported by McDevitt et al. (2003) [107], who used conventional UV light (254 nm). However, conventional UV light (mercury-based) still has higher germicidal efficiency, with respect to electrical energy consumptions, than excimer lamps [114]. Thus, the utilization of excimer lamps as a large-scale germicidal technology remains a subject of future improvements in its energy efficiency.

In another study, far UV-C (222 nm) treatment was found to reduce the viral loads of aerosolized human coronaviruses (HCoV 229E and HCoV OC43; 3–4 log) and also the expression of viral spike glycoproteins (protein for host attachment), as confirmed by median tissue culture infectious dose (TCID_50_) and immunofluorescence assays in human lung cells, respectively [85]. Further, the efficacy of KrCl excimer lamps (222 nm; D_90_ = 0.56 mJ/cm^2^) [85] against coronaviruses was comparable to conventional UV light (254 nm; D_88_ = 0.6 mJ/cm^2^) [115].

### 3.3. Inactivation of Viruses in Liquid Matrix

Several studies have identified the presence of coronaviruses (or their viral RNA) in liquid matrixes, such as in drinking water [116], milk [117], or blood plasma [118]. A human coronavirus (HCoV 229E) was projected to survive in unfiltered tap water for 588 days at 4 °C, although 99.9% of the virus was inactivated within 12.1 days at 23 °C [116]. Moreover, viral RNA of SARS-CoV-2 has been detected in untreated wastewater treatment [27,28], albeit another study found that coronaviruses (HCoV 229E and feline infectious peritonitis virus) survived in wastewater for only 2–3 days [116]. In unpasteurized cow, goat, and camel milk, MERS-CoV survived at 4 °C and to a lower extent at 22 °C, with varying degrees of reduction over 72 h [117]. Further, a review by Chan, Yang, and Wang (2020) listed several studies that had detected viral RNA of SARS-CoV, MERS-CoV, and SARS-CoV-2 in blood plasma of infected individuals (symptomatic and asymptomatic) [118], which prompted WHO [119] and the Food and Drugs Administration (FDA) [120] to draft recommendations on the theoretical risks of coronavirus transmission through transfusion of blood products.

There has been no consensus on the precise mechanism of viral inactivation by UV light, especially with different wavelengths used in different studies. A study found no conclusive correlation between genomic properties (type of nucleic acid and genome size) and light-based inactivation of viruses in liquid matrix and thus, the authors suggested that other factors, such as proportions of the genome that encode essential genes, genome packaging, capsid structure, and presence of an envelope, may also influence the efficacy of light-based inactivation [121]. However, others reported that viral DNA was more prone to UV-light (254 nm) damage in buffer solutions than viral RNA, with dsRNA being the most resistant [122]. Similarly, bacteriophages with ssRNA genomes (MS2 and Qβ) were more resistant to UV-light inactivation (266 and 279 nm) in distilled water than the ssDNA bacteriophage ΦX174 [123] (As shown in Table 2).

A study demonstrated that UV-light (254 nm) inactivation of viruses in animal plasma (bovine and porcine) was more effective for those with an outer envelope than for the non-enveloped viruses (Figure 3), regardless of the viral genome size and type of nucleic acid [138]. However, other authors suggested that the viral envelope of hepatitis C virus was not affected by UV-light (254 nm) treatment, which only targeted the viral genome, and potentially the viral capsid [139]. Thus, the underlying reason for the apparent higher UV susceptibility of enveloped viruses than non-enveloped viruses, especially in the absence of photosensitizing agents, remains undefined.

Other factors that may influence the efficacy of UV-light inactivation include the type of UV light used and the presence of organic matters. While UV-C at 254 and 222 nm have been reported to be potent germicides, a study found that UV-A (365 nm) had no effect on SARS-CoV in liquid suspensions [126]. Further, the presence of proteins may confer protective effects on viruses against light inactivation. Results of two studies showed that viral inactivation by broad white light treatment required a higher dose to inactivate a range of viruses, which included several ssRNA viruses, in phosphate buffered saline (PBS) containing fetal bovine serum (5% *v*/*v*; 0.2% *v/v* protein) than in non-proteinaceous PBS [121,128]. Humic acid (organic acid) could also form a coating layer around the surface of MS2 bacteriophage in water, which protects the virus from UV light [140,141]. However, these organic materials may not be representative of the substances present in human respiratory droplets, which mainly contain salt, mucin glycoprotein (in human airway mucus; 75–90% carbohydrate), and surfactants [142,143]. Pulsed-light technologies provide an alternative approach to rapid inactivation of viruses. For instance, a study found that pulsed-UV (266 nm) laser technology completely inactivated bovine viral diarrhea virus (BVDV; approximately 10^4^ PFU/mL) in less than 1 s, whereas a conventional UV (254 nm) lamp required 30–60 min to achieve the same rate of inactivation [127]. In agreement, other authors found that pulsed-white light technology (200–1100 nm; 2 J/cm^2^ and 1.27 J/cm^2^ per pulse) achieved at least 5-log reduction in viral loads of several ssRNA viruses in a clear liquid matrix within 2 s [121,128]. However, two studies reported lower viral reductions (1–3 log) after three seconds of treatment of similar matrixes by pulsed white light at lower energy intensities (200–1000 or 180–1100 nm; 0.67 J/cm^2^ and 0.27 J/cm^2^ per pulse) and thus demonstrated that virucidal activity may vary with the pulsed-light technology used [129,130].

Several advantages of pulsed broad white light over a conventional UV lamp include higher energy intensity [121], higher efficiency at converting electrical energy into photon energy [128], improved safety (non-mercury) [144], inclusion of a wider range of light spectrum [145], and higher degree of control over the flash duration and frequency output [145]. Although, due to its short exposure time, pulsed-light treatments may not be able to penetrate some matrixes, as apparent from its lower virucidal efficacy in the presence of proteins (Table 2), and thus may only be used as a surface sanitization technology or in transparent media [145]. For turbid liquid media, such as blood platelets, THERAFLEX-UV may be more suitable.

Results of three studies indicated varying viral reductions of hepatitis C virus (5 log), BVDV (2.7 log), MERS-CoV (3.7–3.8 log), Ebola virus (4.5–4.6 log), SARS-CoV (3.4–3.6 log), Crimean-Congo haemorrhagic fever virus (2.2–2.9 log), and Nipah virus (4.3–4.6 log) in blood platelet concentrates after treatment by THERAFLEX-UV [133,134,135] (Table 2), which combines vigorous agitation with UV-light treatment to achieve uniform UV penetration throughout the liquid [146]. This technology involves the loose placement of blood bags between two holders to allow for free movement of the liquids and the subsequent UV-C (254 nm) illumination under agitation (110 rpm) induced by an orbital agitator [146].

Recent developments of UV LED have allowed for a wider coverage of the UV-light spectrum for different applications [147]. These developments include emergences of AlN (aluminium and nitride)-based LED (210 nm) and AlGaN (aluminium, gallium, and nitride)-based LED (222–351 nm) [148,149]. In one study on Qβ bacteriophage (in PBS), UV LED exhibited the highest inactivation rate at 265 nm, relative to other wavelengths (280 and 300 nm), although the most efficient virucidal effect, in terms of energy consumption, was observed at 280 nm—conventional UV lamp (254 nm) still had a higher energy efficiency than all of the UV LEDs tested [131]. Another study reported a 3-log reduction of SARS-CoV-2 (initial titre of 3.7 × 10^4^) in PBS within 10 s of treatment by UV LED (280 nm; 37.5 mJ/cm^2^), with a final viral reduction of 3.33 log (below limit of detection of 1.3 log) achieved after 20 s (75 mJ/cm^2^) [136]. These findings indicate that virucidal UV LED may potentially be used as an alternative germicidal technology to conventional UV lamps, especially due to its wider selection of wavelengths, although further research is required to improve its energy efficiency.

Tailing effect may occur during UV-light inactivation of viruses, i.e., the rate of inactivation decreases at higher UV-light fluences, after an initial exponential decay [150]. The possible explanations for this phenomenon include light shielding due to aggregations of viral particles or the presence of resistant viral sub-populations, albeit no conclusive evidence is available to support these notions [150]. Others argued against those suggestions and proposed that injured viruses with damaged genomes may form aggregates and infect the same host cell post-treatment and thus an intact genome could be constructed from undamaged portions of the individual viral genomes via genomic recombination; these studies were conducted using bacteriophages (MS2 or T1-T7 coliphages) in clear liquid matrixes (buffer solution or diluted phage stock) [151,152].

### 3.4. Inactivation of Viruses on Solid Surfaces

Due to the increased demand for personal protective equipment (PPE) as a result of the current SARS-CoV-2 pandemic, the decontamination and reuse of filtering facepiece respirators (FFR), such as N95, may be required to ensure continued availability [153]. In two studies, influenza A (H1N1) viruses on N95 FFR remained infectious for 6 days [154,155] and thus these findings highlight the need for adequate disinfection procedures to improve the safety of prolonged use of FFR or other similar materials.

On FFR, at least a 4-log viral reduction was reported for enveloped Influenza A viruses (H5N1 and H1N1) and a 3-log reduction for non-enveloped MS2 bacteriophage at UV-light fluences of 1.8 J/cm^2^ and 4.32 J/cm^2^, respectively [156,157,158]. Another study found that treatment of N95 FFR by pulsed-xenon UV technology reduced the viral load of SARS-CoV-2 by 4.79 log within 5 min [159]. However, while viral infectivity was lost after UV-light treatment, the RNA genetic signature may be retained [156,160], which could be undesirable in settings that are prone to environmental contaminations, such as in a molecular diagnostic laboratory. Further, a study also showed that a small portion of SARS-CoV (13.1 PFU/mL of the initial 2.65 × 10^7^ PFU/mL) was able to survive prolonged UV-light treatment (254 nm; 60 min) on a stainless steel bench surface [161]. Thus, the use of ethanol (70%) has been recommended to eliminate this fraction of residual viruses [161] and viral RNA [160] that was resistant to UV-light treatments.

Similar to liquid, inactivation of viruses on solid media is influenced by the presence of proteins as a protective agent. These viruses may also persist on surfaces after UV-light treatment, particularly when dried. A study found that 3% to 10% of ssRNA viruses (Ebola, Vaccinia, and Lassa) dried onto non-porous glass surfaces were able to survive UV-light treatment (254 nm; 110 J/m^2^ in 30 s), which was potentially due to the presence of proteins (fetal calf serum) and cell debris in the growth medium that protected these fractions of the viral population against the UV light [162]. Other authors found that the inactivation of murine norovirus-1 on high-density polyethylene, polyvinyl chloride, and stainless steel was significantly lower in the presence of proteins [128,129]—less reduction in viral load observed at the same light dosage [128] or more than twice the light dosage required to achieve the same viral reduction [129] (Table 3).

The efficacy of germicidal UV is also dependent on the type of application surface, regardless of the delivery modes (continuous or pulsed). Fino and Kniel (2003) reported varying degrees of UV-light (254 nm) inactivation of hepatitis A virus, feline calicivirus, and Aichi virus on lettuces (3.48–4.62 log), green onions (2.35–5.58 log), and strawberries (1.13–2.60 log). The highest level of inactivation was observed in virus-contaminated lettuces due to their smooth surface and lowest inactivation was in strawberries due to their rough surface (protective effect due to seed pockets and shadowing) [163]. Similarly, Belliot et al. (2013) found that pulsed-white light (200–1100 nm) was at least 7 times more effective in inactivating MS2 bacteriophage on glass beads than on complex food matrixes, such as powdered black pepper, garlic, and chopped mint [164].

Finally, the practical application of UV-light sanitization depends on its ability to inactivate viruses over distances beyond those typically used in laboratory-scale experiments. In one study, mouse hepatitis virus A59 (MHV) and MERS-CoV spotted on glass coverslips placed in UV-transparent petri dishes were inactivated when placed at a distance of 1.22 m from the UV-light source (254 nm; whole room UVGI system). The reduction in the viral titre was 6.11 log (10 min) and 5.95 log (5 min) for MHV and MERS-CoV, respectively [165]. However, as upper-room UV-light germicidal devices must be installed at a safe height, the inactivation of coronaviruses at further distances needs to be ascertained in future research.

### 3.5. Comparison of Light Dosage across Studies

Two UV-light inactivation models were used to determine the UV-light dosage required to achieve 90% inactivation of ssRNA viruses (D_90_). Briefly, UV-light inactivation may follow an exponential disinfection model, i.e., n_t_/n_o_ = e^−*k*D^, where n_t_/n_o_ is the viral survival rate, D is the UV-light fluence (J/cm^2^), and *k* is the slope of the survival curve when n_t_/n_o_ is plotted against D [167]. In a linear model, i.e., n_t_/n_o_ = 10^−*k*D^, the inverse of the *k* value (slope of the survival curve) can be defined as the D_90_ value [131,168]. Only studies from the past 20 years that had details on the D_90_ values, *k* values, or the slope of the linear regression equation used were included. Several studies assessed viral inactivation in multiple liquid matrixes, in which case the inactivation rate was taken from the liquid matrix with the least solid content. Results are presented in Table 4.

Walker and Ko (2007) compared their experimental finding on the UV susceptibility of three viruses (MS2 bacteriophage, adenovirus mouse hepatitis coronavirus) in aerosols with data from the literature on the UV susceptibility of the same three viruses in liquid suspension. The authors concluded that these viruses were 7 to 86 times more resistant to UV inactivation in liquid than in aerosols, regardless of their size (90–100 nm or 30–40 nm), type of nucleic acid (RNA or DNA), and viral structure (enveloped or non-enveloped) [115]. This finding is consistent with the data presented in this review (Table 1, Table 2 and Table 4).

There was no relationship between viral genome size and D_90_ from the data evaluated in this review (R^2^ = 0.03 and 0.04 for airborne and liquid inactivation, respectively; calculated from data in Table 4) and this is consistent with the finding of Kowalski et al. (2009) [44]. However, the inactivation rate (D_90_) of the same virus varied across different studies, possibly due to the different viral strains and composition of the liquid matrixes used [169]. Thus, future research is required to establish UV-based inactivation models for different viruses that accommodate potential variations across different application matrixes.

## 4. Virucidal Properties of Blue Light

### 4.1. Mechanism

Blue light (380–500 nm) is known to exhibit germicidal activities, especially within the wavelength range of 400 to 450 nm. For instance, the antimicrobial potency of blue LED has been demonstrated against several pathogenic bacteria, such as *Salmonella* spp. [176,177], *Listeria monocytogenes* [176,177,178,179,180], *E. coli* [181,182], and bacterial endospores (*Bacillus cereus*, *B. subtilis*, and *Clostridium difficile*) [183]. Generally, the mechanism of blue light sanitization is associated with the absorption of the light by photosensitizers, especially the endogenous porphyrin (Figure 4) [35], which can result in the generation of ROS [184]. These ROS may induce damage to proteins, lipids, and nucleic acids [185], which subsequently leads to the loss of cell membrane permeability [186], the leakage of intracellular potassium ion (K^+^) due to denaturation of the transmembrane protein Na^+^/K^+^ pump (facilitated by the oxidation of lipid membrane) [187], and the oxidation of bacterial DNA [188].

Due to the lack of endogenous porphyrin in viral particles, blue-light inactivation of viruses would require the addition of exogenous photosensitizers [189]. In the presence of light, photosensitization may inactivate both enveloped and non-enveloped viruses: positively-charged photosensitizers can induce damage to nucleic acid through the oxidation of guanosine residues, whereas negatively-charged photosensitizers target the viral envelope—aminolipids and peptides can be a potential target, which can lead to the inactivation of membrane enzymes and receptors [190]. Moreover, similar to bacterial cells, a viral lipid envelope may undergo peroxidation (in the presence of photosensitizers), which leads to the loss of membrane integrity and fluidity and also to an increase in membrane permeability [191]. There are still limited data on blue-light inactivation of viruses, although a review by Tomb et al. (2018) suggested that blue light (380–480 nm) had a lower germicidal potency against viruses than against bacteria and fungi [192].

### 4.2. Viral Inactivation

The literature contains no data on the implementation of germicidal blue light against aerosolized microorganisms, with the majority of the studies on viruses being conducted in liquid media. Due to limited data available for ssRNA viruses, we also included studies on non-ssRNA viruses. There are two types of technology used in the studies on virucidal blue light, namely blue LED (continuous) and femtosecond laser (pulsed). There are no published data on the use of pulsed-blue LED against viruses, although the technology has shown bactericidal activity against *Propionibacterium acnes* [193,194] and *S. aureus* [195], with pulsed-blue LED (405 nm) also reported to be more energy efficient than continuous blue LED (405 nm) in inactivating *S. aureus* [195].

As expected, the presence of photosensitizing agents is an important factor in viral inactivation by blue light. The inactivation of feline calicivirus (FCV) in phosphate buffered saline (PBS) required 2804 J/cm^2^ of blue light (405 nm) to achieve a 3.9-log reduction. However, the supplementation of riboflavin, tyrosine, tryptophan, pyridoxine, and folic acid into PBS resulted in a 5.1-log reduction after exposure to only 421 J/cm^2^ of blue light. Further, blue-light inactivation of FCV was also enhanced in organically-rich media: culture medium with fetal bovine serum (4.8 log reduction at 421 J/cm^2^), artificial saliva (5.1 log at 421 J/cm^2^), blood plasma (4.8 log at 561 J/cm^2^), and artificial feces (4.5 log at 1400 J/cm^2^) [196]. Similar results were obtained for bacteriophage ϕC31, with viral inactivation in proteinaceous and porphyrin-rich media being higher than in minimum media—viral counts in treated minimum media was still significantly (*p* < 0.05) lower than untreated controls [197].

Two studies assessed the application of blue light to minimize the spread of viral infectious diseases in aquaculture production systems. Ho et al. (2019) found that blue light (405 nm) treatment completely inactivated viral hemorrhagic septicemia virus (VHSV) in L15 culture medium and flounder fish skin mucus. The survivability of olive flounders in artificially-contaminated rearing water was also higher in the blue-light treated group than in the untreated control group. A possible explanation for the inactivation of VHSV was due to the presence of photosensitizers in L15 medium (riboflavin, tryptophan, tyrosine, pyridoxine, and folic acid), fish skin mucus (mucin), and the rearing water (mucin from the fish skin mucus and fish feces) [198]. Future studies can also be designed to assess the virucidal activity of blue light against respiratory viruses in human airway mucus or similar matrixes.

In another study, exposure to blue LED (470 nm; 3.6 J/cm^2^) in the presence of curcumin resulted in damage to the viral capsid and RNA of murine norovirus-1 [199]. The inactivation of murine norovirus-1 was dependent on the concentration of curcumin, both in live oysters and culture medium, with the maximum viral reduction of 3 log observed at a curcumin concentration of 20 µmol/L. These findings indicated that virucidal activity could also be achieved at longer blue-light wavelengths (>450 nm), in combination with a suitable photosensitizer [199].

The germicidal mechanism of blue light—other than the activation of photosensitizers—is still relatively undefined. As mentioned above, inactivation of FCV and bacteriophage ϕC31 in minimum media (no photosensitizers) had been observed [196,197], which indicates a possibility of alternative inactivation mechanisms. Tomb et al. (2017) [196] proposed two hypotheses to explain this phenomenon: (1) UV-A (390 nm) was generated by the blue LED, which could induce oxidative damage to viral capsids after extended exposure; (2) small amounts of blue light at 420–430 nm emitted from the source may contribute to viral inactivation—the authors referred to a study by Richardson and Porter (2005), who found that prolonged storage of murine leukemia virus in ambient light (420–430 nm; 200 lux for 3 to 7 days) led to a loss of infectivity at a higher rate than the control samples (stored in the dark) [200]. However, Tomb et al. (2017) [196] did not measure the specific dosages of the residual UV-A (390 nm) and blue light (420–430 nm) potentially emitted by the blue LED used and thus, these hypotheses are subjects of further investigations.

Blue light can also be delivered as ultrashort pulses with the use of femtosecond laser, which may disrupt the capsid structures of non-enveloped viruses through a mechanism called Impulsive Stimulated Raman Scattering (ISRS) [201,202,203]. ISRS relies on the excitation of a coherent acoustic Raman-active vibrational mode, which is associated with the vibration of viral capsids [201,202,203,204]. While the blue-light femtosecond laser (425 nm) disrupted the capsid structures of non-enveloped viruses, it left the viral genome relatively intact [202,203]. In contrast, in enveloped viruses, the structures of viral envelope, capsid, and genome remained undisturbed [205]. Instead, the ultrafast laser irradiation (425 nm) induced partial unfolding of the viral proteins by disrupting hydrogen bonds and/or hydrophobic interactions, leading to selective aggregation of the viral capsid and tegument proteins and thus inactivating the virus [205]. Interestingly, femtosecond laser at 400 nm and 800 nm induced damage to both the viral capsid and genome in M13 bacteriophage [204].

In summary, blue light appears to possess some virucidal activity, especially in the presence of photosensitizers. The use of femtosecond laser may circumvent the need for photosensitizers, although improvements in its efficiency are required as femtosecond laser treatments may take up to 10 h [201]. Although blue light is still less energy efficient than UV light in inactivating viruses, especially when compared with the available pulsed-light technologies (Table 2, Table 3 and Table 5), it remains a viable alternative due to its safety. Generally, blue light is considered to be less harmful to mammalian cells than UV irradiation, albeit its photo-toxicity is wavelength-dependent [36]. In addition to safety benefits, prolonged exposure of materials to blue light at 405 nm would induce less photo-degradation than UV irradiation (254–260 nm) [36].

## 5. Virucidal Properties of Red Light

Several studies have reported on the germicidal activity of other light bands beyond the UV and blue light spectra. Across these light regions, red light seems to have the best potential for being developed into a sanitization technology, in combination with the photosensitizer methylene blue (MB). In one study, red light (600–700 nm) reduced the synthesis of oxidative stress proteins in *E. coli* in seawater to a higher degree than exposure to white (400–700 nm), blue (400–500 nm), or green (500–570 nm) lights, but only when MB was present [207]. Similarly, flexible-organic red LED (669–737 nm) exhibited a strong (99% inactivation) germicidal effect on *S. aureus* in the presence of MB (absorption spectrum of 500 to 700 nm; absorption peak at 662 nm) [208]. However, in the absence of MB, red LED (698 nm) exhibited no bactericidal effects on *Pseudomonas fluorescens* and *Staphylococcus epidermidis*—green (570 nm) and yellow (584 nm) lights also showed no effects on these bacteria [209]. Thus, the germicidal activity of red light is dependent on the presence of MB, regardless of the light sources used, such as fluorescence lamp [207] and LED [208,209].

The virucidal properties of red light also seem to be dependent on MB. Floyd, Schneider, and Dittmer (2004) proposed two mechanisms for red light/MB inactivation of RNA viruses: (1) RNA-protein cross-linkage as the primary event; (2) formation of 8-hydroxy-guanosine in RNA, which was strictly dependent on the presence of oxygen, as an accompanying process [210]. As a phenothiazine dye, MB intercalates into viral nucleic acids, with subsequent illumination generating singlet oxygen, which can lead to the photo-oxidation of guanosine and thus to the destruction of viral nucleic acid [133].

A red-light (630 nm) dosage of 30 J/cm^2^ was required to inactivate MERS-CoV (3.3–3.5 log), Ebola virus (4.6–4.7 log), SARS-CoV (3.1–3.2 log), Crimean-Congo haemorrhagic fever virus (3.2 log), and Nipah virus (2.7–3 log) in blood plasma [134,135]. Similarly, hepatitis C virus (3.83 log) and BVDV (3.59 log) in blood plasma were inactivated by red light (630 nm; 40 J/cm^2^) [133]. As already mentioned (Table 2), UV light could also be used to inactivate these viruses in plasma-reduced platelet concentrates [133,134,135] and thus, red light-based and UV-based sanitization technologies could be used synergistically to reduce infectious viruses in blood products.

Results of two studies suggested that red light-based treatment was only effective at reducing the viability of enveloped viruses, but not non-enveloped viruses [211,212]. However, membrane lipid oxidation has not been associated with MB photo-inactivation and thus, the underlying reason for the higher susceptibility to red light/MB treatment of enveloped viruses than non-enveloped viruses remains undefined.

## 6. Photosensitizing Agents for UV-A and UV-B

While UV is a potent germicidal agent, its efficacy is mostly restricted to the UV-C region. Nevertheless, several photosensitizers can absorb UV light at longer wavelengths and thus may be used to enhance the germicidal activity of UV-A and UV-B. For instance, amotosalen, which belongs to the mutagenic chemical group called psoralen, is commonly associated with UV-A (320–400 nm), especially in the treatments of blood plasma [213]. Psoralens can intercalate to nucleic acids (DNA or RNA) and upon radiation by UV-A, become covalently bound to the pyrimidine bases (cytosine, thymine, or uracil), which leads to the formation of crosslinks in the nucleic acids [213].

In several studies, the combination of amotosalen and UV-A was effective against enveloped ssRNA viruses, such as SARS-CoV (5.8-log reduction), MERS-CoV (4.48–4.67 log), human immunodeficiency virus (6.1–6.2 log), human T-lymphocyte tropic virus (4.2–4.6 log), hepatitis C virus (4.5 log), and BVDV (6 log), in blood plasma or platelets [214,215,216]. For non-enveloped viruses, the sensitivity to amotosalen/UV-A treatment varied considerably, with post-treatment reduction in viral titre of feline conjunctivitis virus (ssRNA virus), human adenovirus (dsDNA), and bluetongue virus (dsRNA) ranging from 1.7–2.4 log, 5.2 log to 5.6–5.9 log, respectively [214].

Alternatively, the combination of riboflavin (500 µmol/L) and UV-B (Mirasol PRT system; 280–320 nm) [217] has also shown virucidal activity against MERS-CoV and SARS-CoV-2 [218,219,220]. In studies on human blood plasma, the viral titre of MERS-CoV and SARS-CoV-2 was reduced by 4.07–4.42 log and 3.40–4.79 log, respectively [218,219,220]. Further, SARS-CoV-2 was also inactivated in platelet (4.53 log) and whole blood units (3.30 log) [219,220].

These findings demonstrate that there are available alternatives to UV-C, especially as the use of UV-C has been associated with safety concerns. However, to date, studies on ssRNA viral inactivation by the combination of UV light and photosensitizing agents have been restricted to application in blood products. Thus, there is scope for future studies on the use of photochemical-mediated UV sanitization for viruses in other matrixes, including in air or on solid surfaces. A review by Wiehe et al. (2019) provides discussions on a range of available photosensitizing agents that can be used in antiviral phototherapy [221].

## 7. Hurdle Technology

There are several technologies that can be coupled with light treatment to achieve higher efficacy against viruses in the air or on solid surfaces. Terrier et al. (2009) found that the combination of cold oxygen plasma (COP) and UV-C (254 nm) reduced the viability of aerosolized human parainfluenza-3 (hPIV-3), respiratory syncytial virus (RSV), and influenza virus A (H5N2) by 4, 3.7, and 4 log, respectively. This was more effective as compared with treatments by COP (3.1 log for hPIV; 2 log for RSV; 2.1 log for H5N2) or UV-C (3.5 log for hPIV; 2.1 log for RSV; 2.5 log for H5N2) alone [222].

The results of one study also showed that vacuum-ultraviolet low-pressure mercury lamp (VUV lamp; 95% emission at 254 and 5% at 185 nm) achieved complete inactivation of airborne MS2 bacteriophage (1.7 × 10^3^ PFU/mL) after only 0.125 s, albeit accompanied by the generation of ozone above the acceptable level of 50 ppb [78]. The addition of catalytic frames Pd-TiO2 (2 mm pleated) reduced the ozone level to 30 ppb, but only when the ozone level was below 150 ppb, which limited the VUV treatment to 0.009 s. At this level of treatment, the viral inactivation of MS2 bacteriophage was less than 1 log (60% inactivation), which is biologically insignificant [78]. Thus, future research should aim at improving the efficiency of both the viral inactivation and ozone degradation by VUV lamps and Pd-TiO2 catalyst frames, respectively, and ultimately at incorporating the two techniques into a feasible germicidal technology.

On solid media, viruses may be internalized or dried and thus protected from UV-light treatment. Two studies found that the combination of UV-C (254 nm) and hydrogen peroxide (vaporized or liquid) enhanced viral inactivation of murine norovirus-1 and MS2 bacteriophage on iceberg cabbage, as compared with treatment with UV-C light alone, although complete inactivation was not achieved in either study [223,224]. Results of one study also showed that the combination of UV/TiO2 photocatalytic treatment (254 nm; 2.7 J/cm^2^) and high hydrostatic pressure (HHP; 500 MPa for 5 min) reduced the titre of murine norovirus-1 by 5.5 log (beyond detection limits) in solidified agar matrix (food model), which was higher than the reduction in viral titre observed when UV light (2.38-log reduction), UV/TiO2 photocatalytic (2.68 log), or HHP (3.5 log) were used as single treatments [225].

The combination of different types of LED may be an alternative approach to achieve higher efficiency. As previously mentioned, the combination of UV-A LED (365 nm) and UV-C LED (265 nm) resulted in a higher inactivation of MS2 bacteriophage than individual treatments [132]. Further, Akgün and Ünlütürk (2017) demonstrated that different combinations of UV-C LED (254 nm), UV-B LED (280 nm), UV-A LED (365), and blue LED (405 nm) resulted in varying degrees of inactivation of *E. coli* K12 and polyphenol oxidase enzyme in apple juice [226]. However, there are no data available on the inactivation of human viruses by multiplex LEDs. Thus, this represents a research gap that could be explored in future studies.

## 8. Future Outlook

The development of improved light-based sanitization technologies, especially those with high virucidal and energy efficiencies, should be targeted as a measure to help control the spread of pathogenic viruses in populations. The use of LED (blue, UV, and broad white light) as a virucidal technology, in singles or combinations, is attractive due to its low cost and low energy consumption. Currently, the low efficiency of LED technology in converting electrical energy input into germicidal output remains its main limitation, as compared with conventional germicidal UV lamps, and should be addressed in future studies. Further research is also needed to assess the feasibility of pulsed-light technology for application in complex systems. For instance, studies can be designed to systematically determine the absorbance of pulsed light by different materials and subsequently to correlate these values with viral inactivation.

We also encourage the scientific community to engage in the development of better models for viral inactivation across different matrixes, particularly for ssRNA and enveloped viruses. A modelling study by Kowalski et al. (2009) [44] may serve as a technical foundation that future work can build upon. While studies have been conducted in vitro, there have been limited attempts at in vivo investigations into ssRNA viral inactivation, especially for airborne viruses or those deposited on solid surfaces. Finally, clearer parameters are needed to determine the light susceptibility of different types of viruses: DNA vs. RNA, enveloped vs. non-enveloped, physical sizes, and shape of capsid—there are conflicting data in the literature.

## Figures and Tables

**Figure 1 pathogens-09-00737-f001:**
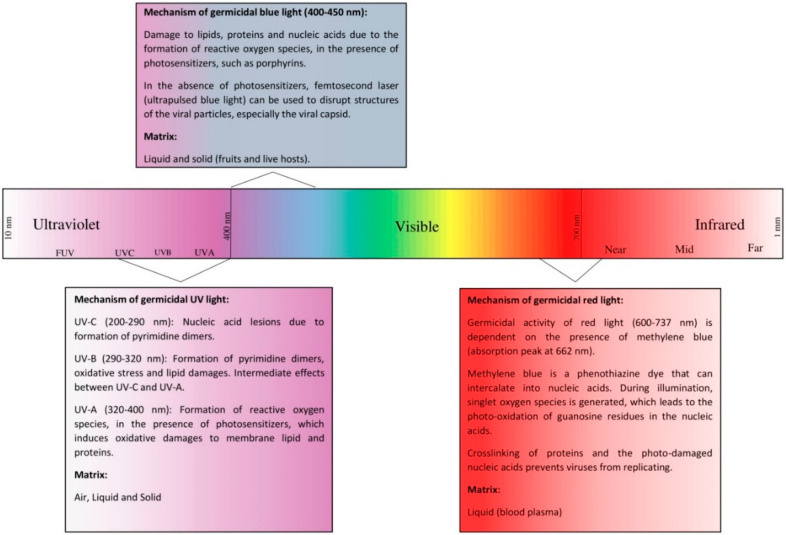
Electromagnetic spectrum and germicidal mechanisms of ultraviolet, blue, and red lights. Listed matrixes for each light represent those for which data are available.

**Figure 2 pathogens-09-00737-f002:**
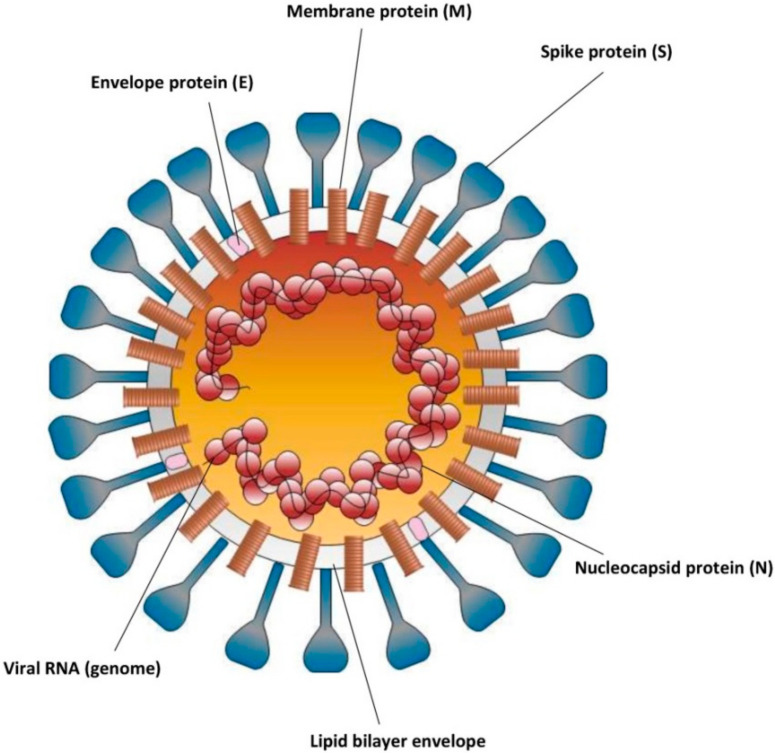
Structure of SARS-CoV-2 virion. The diagram only shows the major structural proteins and viral envelope.

**Figure 3 pathogens-09-00737-f003:**
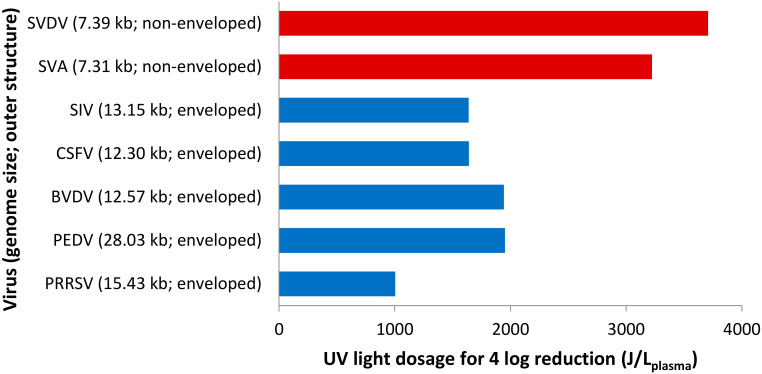
Light dosage required to achieve 4-log reduction of ssRNA viruses in animal plasma by UV light (254 nm). SVDV = swine vesicular disease virus; SVA = senecavirus A; SIV = swine influenza A virus; CSFV = classical swine fever virus; BVDV = bovine viral diarrhea virus; PEDV = porcine epidemic diarrhea virus; PRRSV = porcine reproductive and respiratory syndrome virus. PEDV belongs to the *Coronaviridae* family. The graph was created from data by Blázquez et al. (2019) [138].

**Figure 4 pathogens-09-00737-f004:**
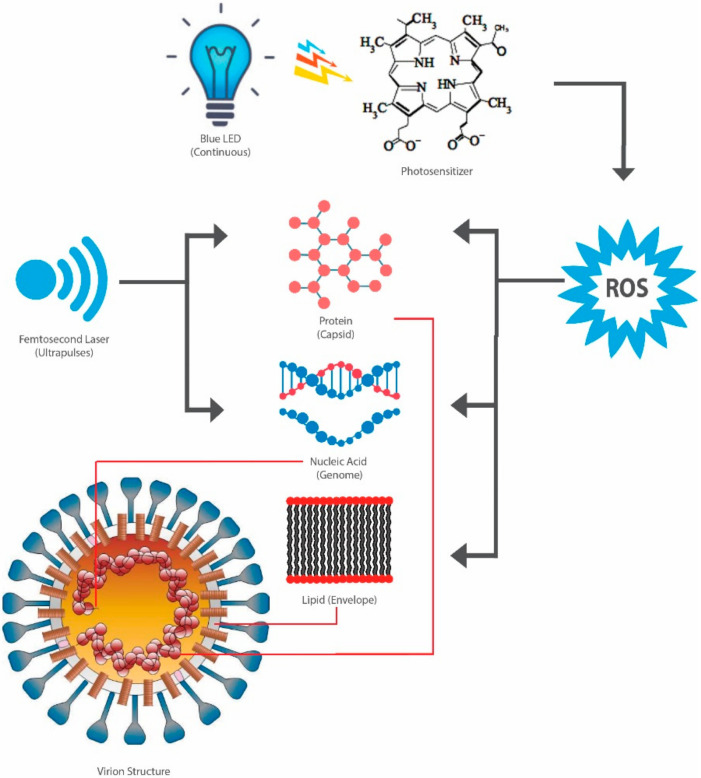
Schematic diagram of blue light viral inactivation pathways: (1) continuous light activates photosensitizers, such as porphyrin, and generates reactive oxygen species (ROS). ROS can attack the viral capsid, genome, and envelope; (2) ultrapulses can be delivered to attack the viral capsid and genome, depending on the wavelength used.

**Table 1 pathogens-09-00737-t001:** UV-based inactivation of airborne ssRNA viruses.

Virus	Initial Titre	Reduction	Light Dosage ^a^	Light Source	Reference
MS2 bacteriophage *	2 × 10^8^ to 7 × 10^8^ plaque-forming units (PFU)/mL	1 log 2 log	0.34–0.42 mJ/cm^2^ 0.80–0.91 mJ/cm^2^	low-pressure, mercury-discharge UV-C lamps (254 nm; 60–253.7 µW/cm^2^)	[97]
MS2 bacteriophage * MHV ^1,^**	10^4^ PFU/mL	<1 log <1 log	2.6 mJ/cm^2^ 0.6 mJ/cm^2^	UV-C lamps (254 nm; 36 W)	[115]
PRRSV ^2,^***	*ca.* 3.33 × 10^6^ TCID_50_/mL	3 log	1.21 mJ/cm^2^	low-pressure, mercury-vapor discharge UV-C lamps (254 nm; 110 V)	[106]
Influenza A ***	5.62 × 10^3^ focus forming units (FFU)/L air	1.4 log	*ca.* 1.48 mJ/cm^2^	low-pressure, mercury UV-C lamps (254 nm; 36 W)	[107]
Influenza A ***	10^8^ FFU/ml	1.3 log	2 mJ/cm^2^	KrCl lamps (222 nm; 120 µW/cm^2^)	[84]
HCoV 229E ** HCoV OC43 **	10^7^ to 10^8^ TCID_50_/mL	3 log 4 log	1.70 mJ/cm^2^ 1.56 mJ/cm^2^	KrCl excimer lamps (222 nm; 0.5–2 mJ/cm^2^)	[85]

Note: * = non-enveloped virus; ** = coronavirus (enveloped); *** = enveloped non-coronavirus; ^1^ MHV = mouse hepatitis virus; ^2^ PRRSV = porcine reproductive and respiratory syndrome virus; ^a^ Maximum light dosage (minimum inactivation rate) was taken, if several dosages proposed under different conditions.

**Table 2 pathogens-09-00737-t002:** UV-based inactivation of ssRNA viruses in liquid matrix.

Virus	Initial Titre	Reduction	Light Dosage ^a^	Light Source	Reference
Sindbis *** HAV ^1,^* EMCV ^2,^*	not available (NA)	2.7 log to >7.2 log (no protein); 1.1 log to 5 log (protein) 3.8 log to >5.7 log (no protein); 1.3 log to >5.6 log (protein) 3.1 log to >5.9 log (no protein); 1.2 log to >6.1 log (protein)	0.25–2 J/cm^2^	Xenon lamp—pulsed white light (200–1100 nm; 300 µs)	[121] ^§^
SARS-CoV **	7 × 10^6^ PFU/mL	6.85 log	0.32 J/cm^2^	UV-C (260 nm; >90 µW/cm^2^; 80 cm; 60 min)	[124]
MS2 bacteriophage * FCV ^3,^*	NA	4 log	0.12 J/cm^2^ 0.04 J/cm^2^	low-pressure, mercury-vapour lamp (254 nm; 8 W)	[125]
SARS-CoV **	10^6^ TCID_50_/mL	4 log	1.45 J/cm^2^	UV-C (254 nm; 4016 µm/cm^2^; 3 cm; 6 min)	[126]
BVDV ^4,^***	10^4.3^ TCID_50_/mL	Complete inactivation (below the limit of detection of 1 log TCID_50_/mL)	1.6 J/cm^2^ (no protein); 3.2 J/cm^2^ (protein) 127 J/cm^2^ (no protein); 190.5 J/cm^2^ (protein)	low-pressure, mercury lamp (254 nm; 40 W) nanosecond laser (266 nm; 8 ns per pulse)	[127]
HAV ^1,^* Murine NoV-1 ^5,^*	10^6^ PFU/mL	2.8 log to 4.8 log (no protein); 2.1 to 2.4 log (protein) 5 log (no protein); 2.8 to 4.1 log (protein)	0.05 to 0.15 J/cm^2^	xenon lamp—pulsed white light (200 to 1100 nm; 2 s)	[128] ^§^
Murine NoV-1 ^5,^*	10^5^ PFU/mL	3.3 log	2.07 J/cm^2^	pulsed white light (200 to 1000 nm; 2 s)	[129]^ §^
Murine NoV-1 ^5,^* Tulane virus *	10^6^ PFU/mL 10^6^ PFU/mL	1.3 log to 5.8 log 1 log to 6 log	0.31 to 4.94 J/cm^2^	pulsed broad white light (180–1100 nm; 40% UV; 6 s)	[130]
Qβ bacteriophage *	NA	3 log	0.03 to 0.5 J/cm^2^	UV LED (265, 280 and 300 nm; 0.99 and 1.01 mW/cm^2^)	[131]
MS2 bacteriophage *	10^6^ PFU/mL	*ca.* 1.5 log	0.02 J/cm^2^	UV LED (265 nm; 10 mW)	[132]
HCV ^6,^*** BVDV ^4,^***	10^5.61^ TCID_50_/mL 10^6.25^ TCID_50_/mL	5 log 2.7 log	0.2 J/cm^2^ 0.2 J/cm^2^	THERAFLEX UV-Platelets system (254 nm)	[133]
MERS-CoV ** Ebola virus ***	10^6.37^ to 10^6.43^ TCID_50_/mL 10^6.84^ to 10^6.96^ TCID_50_/mL	3.7 log to 3.8 log 4.5 log to 4.6 log	0.15 J/cm^2^ 0.15 J/cm^2^	THERAFLEX UV-Platelets system (254 nm)	[134]
SARS-CoV ** CCHFV ^7,^*** Nipah virus ***	10^5.8^ to 10^6^ TCID_50_/mL 10^4.6^ to 10^5.2^ TCID_50_/mL 10^6.2^ to 10^6.5^ TCID_50_/mL	3.4 log to 3.6 log 2.2 log to 2.9 log 4.3 log to 4.6 log	0.15 J/cm^2^ 0.15 J/cm^2^0.15 J/cm^2^	THERAFLEX UV-Platelets system (254 nm)	[135]
SARS-CoV-2 **	3.7 × 10^4^ PFU/mL	3.33 log	75 mJ/cm^2^	Deep UV LED (280 nm)	[136]
SARS-CoV-2 **	NA	*ca.* 5.7 log	16 mJ/cm^2^	low-pressure, mercury lamp (254 nm; 1.08 mW/cm^2^)	[137]

Note: * = non-enveloped virus; ** = coronavirus (enveloped); *** = enveloped non-coronavirus; ^1^ HAV = hepatitis A virus; ^2^ EMCV = encephalomyocarditis virus; ^3^ FCV = feline calicivirus; ^4^ BVDV = bovine viral diarrhea virus; ^5^ NoV = norovirus; ^6^ HCV = hepatitis C virus; ^7^ CCHFV = Crimean-Congo haemorrhagic fever virus; ^a^ Maximum light dosage (minimum inactivation rate) was taken, if several dosages proposed under different conditions; ^§^ Broad spectrum white light (spectrums equivalent to sunlight) used, which included UV light in combination with other electromagnetic wavelengths.

**Table 3 pathogens-09-00737-t003:** UV-based inactivation of ssRNA viruses on solid surfaces.

Virus	Initial Titre	Reduction	Light Dosage ^a^	Light Source	Reference
SARS-CoV **	2.65 × 10^7^ PFU/mL	4 log	0.12 J/cm^2^	UV-C (254 nm; 134 µW/cm^2^)	[161]
MS2 bacteriophage *	10^8^ PFU/mL	1 log	3.20 mJ/cm^2^	UV-C lamp (254 nm; 60–240 µW/cm^2^; 3 s to 6 min)	[166]
HAV ^1,^* FCV ^2,^* Aichi virus *	6.93 × 10^6^ to 6.93 × 10^8^ PFU/mL	2.44 log to 5.42 log 2.12 log to 4.46 log 1.71 log to 4.43 log	0.24 J/cm^2^	low pressure lamp (254 nm)	[163]
MS2 bacteriophage *	10^7^ PFU/mL	3 log to 4 log (below detection limit of 1000 PFU/mL)	4.32 to 7.20 J/cm^2^	low-pressure, mercury-arc lamp (254 nm; 40 W)	[158]
HAV ^1,^* Murine NoV-1 ^3,^*	10^5^ PFU/mL	5 log 5 log (no protein); 2.3 log to 3.6 log (protein)	0.09 J/cm^2^ 0.06 J/cm^2^	xenon lamp—pulsed white light (200 to 1100 nm; 2–3 s)	[128] ^§^
Influenza A virus (H1N1) ***	6.98 × 10^7^ PFU/mL	4.08 log to 5.75 log (PD ^4^ = 15 µm); 4.83 log to 5.08 log (PD ^4^ = 0.8 µm)	1.8 J/cm^2^	UV-C lamps (254 nm; 80 W)	[157]
Influenza A virus (H5N1) ***	2.20 × 10^5^ PFU/mL	4.5 log	1.8 J/cm^2^	UV-C lamps (254 nm; 15 W)	[156]
MS2 bacteriophage *	6.31 × 10^7^ PFU/mL	4.87 log (glass beads); 0.64 log (powdered black pepper); 0.12 log (garlic); 0.68 log (chopped mint)	9.6 J/cm^2^	pulsed white light (200–1100 nm; 0.96 J/cm^2^ per pulse; 1 s)	[164] ^§^
Murine NoV-1 ^3,^*	10^5^ PFU/mL	3 log (no protein); 2.6 log to 3 log (protein)	3.45 J/cm^2^ (no protein); 7.60 J/cm^2^ (protein)	pulsed white light (200 to 1000 nm; 2 to 6 s)	[129]

Note: * = non-enveloped virus; ** = coronavirus (enveloped); *** = enveloped non-coronavirus. ^1^ HAV = hepatitis A virus; ^2^ FCV = feline calicivirus; ^3^ NoV = norovirus; ^4^ PD = particle diameter; ^a^ Maximum light dosage (minimum inactivation rate) was taken, if several dosages proposed under different conditions. ^§^ Broad spectrum white light (spectrums equivalent to sunlight) used, which included UV light in combination with other electromagnetic wavelengths.

**Table 4 pathogens-09-00737-t004:** Estimation of UV-light dosage required for 90% inactivation (D_90_) of ssRNA viruses. All measurements were taken at 254 nm, unless stated.

Virus	D_90_ (mJ/cm^2^)	Genome Size (kb)	Matrix	Reference
MS2 bacteriophage	0.28	3.57	Airborne	[97]
Influenza A (H1N1)	1.04	13.50	Airborne	[107]
Influenza A (H1N1)	1.28 (222 nm)	13.50	Airborne	[84]
Human coronavirus 229E	0.56 (222 nm)	27.32	Airborne	[85]
Human coronavirus OC43	0.39 (222 nm)	30.74	Airborne	[85]
Feline calicivirus	6	7.58	Liquid	[125]
Feline calicivirus	47.85	7.68	Liquid	[168]
Feline calicivirus	5	7.68	Liquid	[169]
Hepatitis A virus	36.50	7.48	Liquid	[168]
Hepatitis A virus	4.5	7.48	Liquid	[170]
Sindbis	10	11.7	Liquid	[170]
Poliovirus-1	24.10	7.44	Liquid	[168]
Poliovirus-1	9.04	7.44	Liquid	[171]
MS2 bacteriophage	22.22	3.57	Liquid	[169]
MS2 bacteriophage	24.39	3.57	Liquid	[172]
MS2 bacteriophage	23	3.57	Liquid	[125]
MS2 bacteriophage	23.04	3.57	Liquid	[168]
MS2 bacteriophage	25.71	3.57	Liquid	[171]
MS2 bacteriophage	18.60	3.57	Liquid	[173]
Qβ bacteriophage	16.81	4.16	Liquid	[171]
Qβ bacteriophage	11.75	4.16	Liquid	[131]
Qβ bacteriophage	10.20 (265 nm)	4.16	Liquid	[131]
Qβ bacteriophage	17.86 (280 nm)	4.16	Liquid	[131]
Qβ bacteriophage	166.67 (300 nm)	4.16	Liquid	[131]
Murine norovirus-1	10	7.38	Liquid	[174]
Influenza A (H1N1)	6400 (365 nm)	13.50	Liquid	[175]
Influenza A (H1N1)	260 (310 nm)	13.50	Liquid	[175]
Influenza A (H1N1)	28 (280 nm)	13.50	Liquid	[175]
MS2 bacteriophage	3.20	3.57	Semi-solid (gelatin-based medium)	[166]

**Table 5 pathogens-09-00737-t005:** Blue light inactivation of viruses.

Virus	Initial Titre	Reduction	Light Dosage ^a^	Light Source	Matrix	Reference
Bacteriophage ϕC31 ^α,^*	10^3^, 10^5^, 10^7^ PFU/mL	0.3 log (M); 2.7 to 7.1 log (NR); 3 log (PR)	306.2 J/cm^2^ (M); 306.2 to 1400 J/cm^2^ (NR); 612.4 J/cm^2^ (PR)	blue light LED (405 nm; 2 cm 56.7 mW/cm^2^)	minimum (M), nutrient-rich (NR) and porphyrin-rich (PR) media	[197]
Murine NoV-1 ^4,^*	*ca.* 10^8.5^ PFU/mL *ca.* 10^4^ PFU/mL	1.32 log (5 µm curcumin); >3 log (20 µm curcumin) 0.75 log (10 µm curcumin); 1.15 log (20 µm curcumin)	3.6 J/cm^2^	blue LED (470 nm; 0.06 W/cm^2^; 60s)	culture medium (5 µm, 10 µm, and 20 µm curcumin) oyster (5 µm, 10 µm, and 20 µm curcumin)	[199]
FCV ^2,^*	2 × 10^5^ PFU/mL	3.9 log (minimum); 4.5 to 5.1 log (ORM)	2804 J/cm^2^ (M); 421 to 1400 J/cm^2^ (ORM)	blue light LED (405; 4 cm; 155.8 mW/cm^2^)	minimum (M) and organically-rich media (ORM)	[196]
Tulane virus *	4 × 10^7^ PFU/mL	0.06 log (no PS ^3^); 0.51 to 1.01 log (PS ^3^)	7.56 J/cm^2^	blue light LED (405 nm; 7.5 cm; 4.2 mW/cm^2^)	blueberries	[206]
VHSV ^1,^***	10^5^ (L15); 10^5.8^ (mucus) PFU/mL	Complete inactivation	715 J/cm^2^	blue light LED (405 nm)	culture medium (L15); Flounder skin mucus	[198]
M13 bacteriophage ^β,^*	10^3^ PFU/mL	Complete inactivation	≥50 MW/cm^2^	femtosecond laser; 425 nm; 40 mW; 10 h	water	[201]
MCMV ^5,α,^***	2 × 10^7^ PFU/mL	5 log	810 J	femtosecond laser; 425 nm; 150 mW; 1.5 h	culture medium	[205]
Murine NoV-1 ^4,^*	5 × 10^7^ PFU/mL	3 log	≥80 MW/cm^2^	femtosecond laser; 425 nm; 120 mWs; 2 h	culture medium	[202]

Note: * = non-enveloped virus; ** = coronavirus (enveloped); *** = enveloped non-coronavirus; ^α^ = double-stranded DNA (dsDNA) virus; ^β^ = single-stranded DNA (ssDNA) virus; ^1^ VHSV = viral hemorrhagic septicemia virus; ^2^ FCV = feline calicivirus; ^3^ PS = photosensitizers; ^4^ NoV = norovirus; ^5^ MCMV = murine cytomegalovirus; ^a^ Maximum light dosage (minimum inactivation rate) was taken, if several dosages proposed under different conditions.

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
