# Peer review of "Control Measures for SARS-CoV-2: A Review on Light-Based Inactivation of Single-Stranded RNA Viruses"

_pathogens, 2020, doi:10.3390/pathogens9090737_

Round 1

Reviewer 1 Report

This review describes the current literature related to the use of different types of lights to inactivate viruses. The authors described the mechanisms of action of the different lights. The manuscript is well written and fluent.

Please address the following concerns:

  1. Tables 1, 2, and 3 do not include percentage of reduction of the viral load. This is a misleading dimension because percentages cannot be lower than 2 folds.
  2. L303, please use consistency in the v/v, either use italics or regular fonts.
  3. L428, use B. subtilis.

Author Response

Reviewer 1

Reviewer’s Comment:

This review describes the current literature related to the use of different types of lights to inactivate viruses. The authors described the mechanisms of action of the different lights. The manuscript is well written and fluent.

Authors’ response:

The authors wish to thank the reviewer for their constructive comments, and the suggestions from the reviewer are addressed as follows:

  1. Tables 1, 2, and 3 do not include percentage of reduction of the viral load. This is a misleading dimension because percentages cannot be lower than 2 folds.

The percentages were removed from Tables 1, 2 and 3, and thus the viral reductions are only presented as number of log reduction.

  1. L303, please use consistency in the v/v, either use italics or regular fonts.

Regular font was chosen and the manuscript was amended (Line 316).

  1. L428, use B. subtilis.

Bacillus subtilis was changed to B. subtilis. In the same line, B. cereus was changed to Bacillus cereus, as this is the first mention of the bacteria in the manuscript (Line 448).

Reviewer 2 Report

In “Control measures for SARS-CoV-2: a review on light-based inactivation of single-stranded RNA viruses” Hadi et al. provide a comprehensive overview of the literature on light-based inactivation of ssRNA viruses, like the virus that causes COVID-19. This work is particularly timely and important given the current pandemic, and is likely to be useful for both scientific and lay audiences. The authors discuss various modes of light-based inactivation, including different wavelengths, sources, and practical applications. The authors also compare inactivation efficiencies in various environments, such as liquids, air, and solid surfaces, which are important for both clinical and public health applications. This review article is beautifully written and the figures enhance the overall clarity of the manuscript. A few minor suggestions are outlined below.

Comments:

  1. Lines 215-233. The authors discuss the impact of relative humidity (RH) on virus inactivation. Relatively recent work has shown that the presence of human airway mucus can modulate the viability of aerosolized viruses (and viruses on solid surfaces) exposed to a range of RH conditions (Yang et al., 2012, PLOS ONE, Kormuth et al., 2018, Journal of Infectious Diseases, Kormuth et al., 2019, mSphere). A factor that has not been well-studied is the role of airway mucus in the stabilization of viruses against UV or other light-based inactivation strategies, which the authors should comment on in this section. Similarly, the authors do well to discuss the role of organic matters, specifically fetal bovine serum (FBS), in virus stabilization in liquids later in the manuscript (Lines 297-305). However, FBS may not be a good surrogate for human airway mucus, and this clarification should be addressed in the manuscript.

  1. In section 3.4, the authors discuss inactivation of viruses on solid surfaces, including N95 respirators. Influenza viruses, been found to remain infectious for days on N95 respirator material (Coulliette et al., 2013, Applied and Environmental Microbiology). This or a similar reference could be mentioned to underscore the need for appropriate inactivation procedures prior to reuse of PPE or other similar materials.

Author Response

Reviewer 2

Reviewer’s comments:

In “Control measures for SARS-CoV-2: a review on light-based inactivation of single-stranded RNA viruses” Hadi et al. provide a comprehensive overview of the literature on light-based inactivation of ssRNA viruses, like the virus that causes COVID-19. This work is particularly timely and important given the current pandemic, and is likely to be useful for both scientific and lay audiences. The authors discuss various modes of light-based inactivation, including different wavelengths, sources, and practical applications. The authors also compare inactivation efficiencies in various environments, such as liquids, air, and solid surfaces, which are important for both clinical and public health applications. This review article is beautifully written and the figures enhance the overall clarity of the manuscript. A few minor suggestions are outlined below.

Authors’ response:

The authors wish to thank the reviewer for their constructive comments, and for providing additional references and discussions to be incorporated into the manuscript. The two suggestions from the reviewer are addressed as follows: 

  1. Lines 215-233. The authors discuss the impact of relative humidity (RH) on virus inactivation. Relatively recent work has shown that the presence of human airway mucus can modulate the viability of aerosolized viruses (and viruses on solid surfaces) exposed to a range of RH conditions (Yang et al., 2012, PLOS ONE, Kormuth et al., 2018, Journal of Infectious Diseases, Kormuth et al., 2019, mSphere). A factor that has not been well-studied is the role of airway mucus in the stabilization of viruses against UV or other light-based inactivation strategies, which the authors should comment on in this section. Similarly, the authors do well to discuss the role of organic matters, specifically fetal bovine serum (FBS), in virus stabilization in liquids later in the manuscript (Lines 297-305). However, FBS may not be a good surrogate for human airway mucus, and this clarification should be addressed in the manuscript.

There are no data available on light-based viral inactivation in human respiratory droplets that contain airway mucus. The three studies mentioned by the reviewer were added into the manuscript and a comment was made to highlight the possibility of future studies on UV-light inactivation in matrixes that resemble respiratory droplets produced during a viral infection (Lines 227-233).

The reviewer is correct to point out that organic materials discussed in the manuscript may not be representative of the matters present in human respiratory droplets. Thus, two studies by Vejerano et al. (2018) and Williams et al. (2006) were added to describe the major components of these respiratory droplets, such as salt, mucin glycoproteins (in human airway mucus) and surfactants (Lines 319-321).

In section 4.2., a study by Ho et al. (2019) assessed the blue-light viral inactivation in fish mucus. Thus, a sentence was added to Lines 500-501 to encourage future studies to investigate the virucidal activity of blue light against respiratory viruses in human airway mucus.   

  1. In section 3.4, the authors discuss inactivation of viruses on solid surfaces, including N95 respirators. Influenza viruses, been found to remain infectious for days on N95 respirator material (Coulliette et al., 2013, Applied and Environmental Microbiology). This or a similar reference could be mentioned to underscore the need for appropriate inactivation procedures prior to reuse of PPE or other similar materials.

Two studies by Coulliette et al. (2013 & 2014) were incorporated into the manuscript. The sentence “In two studies, influenza A (H1N1) viruses on N95 FFR remained infectious for 6 days, and thus these findings highlight the need for adequate disinfection procedures to improve the safety of prolonged use of FFR or other similar materials” was added to Lines 369-372.

Reviewer 3 Report

Hadi et al provide a very clear review on light-based methods of virus inactivation. Overall, the information is timely and useful. It also does a nice job of summarizing what additional studies are needed to further advance light-based disinfection. I noted a few minor issues that, if addressed, might improve the manuscript.

  1. The title says the article is about SARS-CoV-2 and ssRNA viruses. However, it is really an article that broadly addresses light-based inactivation of a variety of viruses and bacteriophages.
  2. Line 51. This paragraph discusses the potential contamination of environmental surfaces by SARS-CoV-2. It would be useful to clarify how little is known about survival of the virus in infectious form on different surfaces and in different matrices under varying environmental conditions.
  3. The paragraphs beginning on line 297 are within the “liquid matrix” section. However, it is not clear that all of the studies described in this section refer to inactivation in liquids. If they do refer to studies on liquid matrices, then it is unclear what liquids were used in several of the cited studies.
  4. Line 323. The authors should explain/define THERAFLEX-UV.

Author Response

Reviewer 3

Reviewer’s comment:

Hadi et al provide a very clear review on light-based methods of virus inactivation. Overall, the information is timely and useful. It also does a nice job of summarizing what additional studies are needed to further advance light-based disinfection. I noted a few minor issues that, if addressed, might improve the manuscript.

Authors’ response:

The authors wish to thank the reviewer for their constructive comments. The four suggestions from the reviewer are addressed as follows:

  1. The title says the article is about SARS-CoV-2 and ssRNA viruses. However, it is really an article that broadly addresses light-based inactivation of a variety of viruses and bacteriophages.

The addition of other viruses was to give context to the limited data available on SARS-CoV-2 and ssRNA viruses.

The major discussions presented in the review concerned the inactivation of ssRNA viruses in different matrixes by currently available light-based technologies. While several non-ssRNA viruses (or even bacteria) were mentioned, this was done only to highlight the mechanisms of light-based inactivation (for e.g. Lines 168-180) and factors that may need to be considered, but no detailed analyses were included for non-ssRNA viruses (for e.g. light dosages, log reductions or available technologies against non-ssRNA viruses). The only exception had to be made for blue light-based inactivation (Section 4.2.) due to the lack of data for ssRNA viruses (three non-ssRNA viruses were included), although the main discussions in this section also focused on ssRNA viruses.  

Besides animal viruses, bacteriophages (ssRNA) were also included to provide extra insights into light-based inactivation due to their common use as surrogates (for e.g. MS2 bacteriophage).Thus, the title specifically mentioned ssRNA viruses and SARS-CoV-2 to inform on the primary group of viruses that the review focused on.   

  1. Line 51. This paragraph discusses the potential contamination of environmental surfaces by SARS-CoV-2. It would be useful to clarify how little is known about survival of the virus in infectious form on different surfaces and in different matrices under varying environmental conditions.

This is a very insightful suggestion indeed and as suggested we have clarified this point in the text by adding a recent review by Aboubakr et al. (2020) (Line 68-71). Aboubakr (2020) summarized the current literature on the persistence of coronaviruses, including SARS-CoV-2, in different matrixes (air, liquid and solid) and under varying environmental conditions (temperature and relative humidity). We believe that to understand the issue suggested by the reviewer, future readers will benefit more by reading the review by Aboubakr et al. (2020) and therefore outside the scope of this review. Also, in Lines 193-197, we already mentioned that the presence of infectious SARS-CoV-2 in the air had only been confirmed in vitro and the epidemiology of the virus was undefined.

  1. The paragraphs beginning on line 297 are within the “liquid matrix” section. However, it is not clear that all of the studies described in this section refer to inactivation in liquids. If they do refer to studies on liquid matrices, then it is unclear what liquids were used in several of the cited studies.

All studies on UV-light viral inactivation mentioned in this section were in liquid matrixes. Clarifications were made as follows:

  1. Liquid matrix (PBS) specified in Line 354
  2. Liquid matrix (PBS) specified in Line 358
  3. Liquid matrixes (buffer solution and diluted phage stock) specified in Lines 371-372

  1. Line 323. The authors should explain/define THERAFLEX-UV.

Brief description of THERAFLEX-UV was added into Line 344: “This technology involves the loose placement of blood bags between two holders to allow for free movement of the liquids, and the subsequent UV-C (254 nm) illumination under agitation (110 rpm) induced by an orbital agitator”. In Lines 347-350, it is already mentioned that this agitation allows for uniform penetration of the UV throughout the liquids.